# CMS: A Continuous Machine-Learning and Serving Platform for Industrial Big Data

**KeDi Li [1] and Ning Gui [2],***

[1] School of Information Science and Technology, Zhejiang Sci-Tech University, Hangzhou 310018, China; leekedi94@gmail.com

[2] School of Computer Science and Engineering, Central South University, Changsha 410083, China

* Correspondence: ninggui@csu.edu.cn

**Abstract:** The life-long monitoring and analysis for complex industrial equipment demands a continuously evolvable machine-learning platform. The machine-learning model must be quickly regenerated and updated. This demands the careful orchestration of trainers for model generation and modelets for model serving without the interruption of normal operations. This paper proposes a container-based Continuous Machine-Learning and Serving (CMS) platform. By designing out-of-the-box common architecture for trainers and modelets, it simplifies the model training and deployment process with minimal human interference. An orchestrator is proposed to manage the trainer's execution and enables the model updating without interrupting the online operation of model serving. CMS has been deployed in a 1000 MW thermal power plant for about five months. The system running results show that the accuracy of eight models remains at a good level even when they experience major renovations. Moreover, CMS proved to be a resource-efficient, effective resource isolation and seamless model switching with little overhead.

**Keywords:** machine-learning; continuous training; industrial big data; container virtualization

## 1. Introduction

With the rapid advancement of Internet of Things and industrial automation systems, enterprises and industries are collecting and accumulating data with unparalleled speed and volume [1–3]. Cloud computing, with capabilities of performing massive-scale and complex computing tasks, has been widely treated as an excellent candidate to meet the growing data processing and computation demands of enterprises and industries [4,5]. Compared with data in other sectors (e.g., government, financial services and retail), industrial data have several distinct characteristics: firstly, the major parts of their data are produced from sensor networks and must be processed quickly [6]; secondly, use of these data is often limited within certain domains for the sake of data safety and regulations. These unique requirements sometimes make traditional cloud computing infeasible since data are often required to upload to the cloud, which causes both latency and security problems.

Compared to cloud computing, the frameworks that process data at the network edge, e.g., fog computing [7,8] and edge computing [9,10], better fit the requirements of industrial big data businesses. Both models push intelligence and processing capabilities down closer to where data originate from. Such design provides a quicker response, more processing efficiencies, less pressure on the network and higher security. On the basic of this model, two major types of services are identified: (1) *trainer*: building models from historical data using machine-learning algorithms for prediction, control or classification tasks. (2) *modelet*: collecting data from sensors, making predictions and control decisions with a trained model for the on-line model serving, similar to the concept of Servable defined in the TensorFlow Serving [11]. Several existing big data platforms, e.g., TFX [12] and IBDP [6],

have already provided certain supports for both types of service. This intelligent system uses data analysis technology and model-based systems to help decision-makers to improve the effectiveness of their decisions. It has a vast area of application including energy, manufacturing industry and medical activity [13,14].

However, in addition to the two major types of requirements, the industrial machine-learning and serving platform has other distinctive requirements. Many industrial processes are mission-critical and might evolve during their 20- to 50-year lives. During their lifecycle, the monitored data and their models evolve over time. For long-time and complex industrial systems, the benefits of accurate modeling are only possible when models are periodically retrained, updated and switched with the latest data. For the production-level deployment, such a type of automation is non-trivial, especially when complications, such as resorce isolation, scheduling, reliability and scalability, need to be taken into account.

This paper proposes a container-based computing platform to automate the workflow from model (re-) learning to deployment, namely CMS. A system component, namely *orchestrator*, is introduced to orchestrate the model training executions across multiple *trainers*. It also provides model deployments and model update services, while keeping the same model-serving processes uninterrupted. The primary goal is to simplify and automate model generation, deployment and switching process. The contributions of the paper are summarized as follows:

- A container-based computing platform is constructed for continuous model learning and serving;
- A model management service, *Orchestrator*, is proposed to streamline the model updating process. The structures for *trainer* and *modelet* are also formulized to ensure consistent invocation and avoid ad-hoc glue codes;
- A continuous model switching mechanism is proposed. It allows the model serving process to be uninterrupted even the model of the *modelet* is updated.

A prototype of the platform is implemented on the top of Rancher® and the open source Docker [15] for service resource isolation, scheduling, service discovery and health checks. Several machine-learning suits, e.g., Spark ML, TensorFlow® and Keras®, are supported for both learning and serving by a custom-designed serving engine. A prototype is developed for a 1000 MW thermal power station power plant with eight different models. More than five months' continuous service shows the necessity of the autonomous model updating and system's feasibility and stabilities. Experimental results show it also exhibits good performance upon three tested scenes.

The rest of the paper is organized as follows. Section 2 introduces the existing literature on industrial computing platforms for big data. In Section 3, the architecture of CMS is introduced and the key components are illustrated. Section 4 introduces the Orchestrator, the structure of trainers and the seamless model updating mechanism. Section 5 presents a case study and experimental results with respect to different system metrics. Section 6 concludes this work.

## 2. Related Work

In a recent review paper, Chen et al. [16] focused on the four phases of the value chain of big data, i.e., data generation, data acquisition, data storage, and data analysis. The discussions are mainly focused on the works for the data analysis stage: both online model processing and the off-line model generation.

### 2.1. Computing Platform for Industrial Big Data

Reference [17] discussed the essential features of cloud computing with regard to the industrial enterprises and pointed out that the computing should be distributed across different domains.

In order to better support the localized data processing, Chang et al. [18] present a platform that addresses edge computing specific issues by augmenting the traditional data center with service nodes placed at the network edges. Hao et al. [19] discuss the software architecture design for fog

computing, which can incorporate different design choices and user-specified polices. Liu et al. [20] review the recent big data platforms that provides online processing capabilities. Reference [6] and [21] propose a similar architecture for industrial big data that incorporate data storage, stream processing and supporting architecture with both active maintenance and the offline prediction and analysis. Tseng et al. [22] discuss the demands for industrial service scalability and present a fuzzy-based, real-time auto-scaling mechanism for resource dynamic allocation. More recently, Geng et al. propose an enhanced industrial big data platform in order to reduce the data processing time while requiring less data storage space by choosing the best data compression and serialization methods [23].

In terms of platform implementation, B. I. Ismail et al. [24] evaluate the Docker container technique for industrial big data. Their results show that Docker provides fast deployment, small footprint and good performance, which make it potentially a viable edge computing platform.

### 2.2. Learning & Serving for Industrial Big Data

There exist many general machine-learning solutions, which can be applied in the industrial big data platform, e.g., TensorFlow, Spark ML. They provide the different machine learning supports. However, as pointed out in [25], the algorithms only represent a small fraction of all the demands of a machine learning platform. It is important to provide distributed computation, scheduling, isolation supports that might be beyond the capabilities provided by single-machine solutions [26]. Furthermore, it is important to manage the different versions of models [27], as models would normally be updated periodically.

A few recent researches identify the need to provide a flexible, high-performance serving system for production environments. Shahoud et al. [28] propose a generic framework to facilitate training, managing, and storing ML models in the context of big data by using an easy-to-use web interface which hides the complexity of the underlying runtime environment from the user. TensorFlow Serving [11] makes it easy to deploy new algorithms and experiments, and supports the updating of models. However, the model regeneration and deployment processes still largely rely on experts. MArk (Model Ark) is a scalable, cost-effective model serving system in the public cloud. Their focus is mainly on the response time of services, without considering the model lifecycle [29]. TFX [12] from Google, identifies the importance of simplifying model training and deployment process, and reduces the production time from months to weeks. However, the model training process and deployment still rely on human-in-loop and fail to provide autonomous model updating support.

## 3. Platform Architecture

In this section, the requirements for CMS are first analyzed, especially for long-time analytics and serving. Subsequently, a container-based architecture is designed and the key components are illustrated.

### 3.1. Requirement Analysis

- **One platform for learning and deployment**. For an industrial machine-learning platform, it is important to support both an off-line *trainer* for model generation from historical data and online model serving for real-time prediction. For the off-line *trainer*, several machine learning suits, e.g., Spark ML, TensorFlow® and Keras®, should be supported as they are widely used by different developers. The different machine learning suits adopt different model formats, e.g., the HDF5 format for Keras models and the TensorFlow specific model format. The supports for the different machine learning suits mean that the model serving implementations should be designed independently with different model formats. Thus, the current main stream approaches should be seamlessly integrated. The compatibility of the existing modeling efforts is quite important for many existing industrial environments in which the model designs normally need considerable work from both data scientists and industrial domain-specific experts;

- **Autonomous model re-training**. Due to the strong time series nature of industrial data, the obtained target value error may become larger as time goes on, leading to the loss of reference value for the target value. For an industrial model, the machine learning pipelines are carefully pre-defined via machine learning experts. The training processes execute the feature selection, feature construction and training process in a defined sequence. When the newer data arrive, it is important to update the model by executing the same computing algorithm again with data within a certain range. By doing so, an up-to-date model with a newer version number is generated and needs to be deployed into the model serving process. As the machine-learning platform is deployed close to the data source and it is mainly maintained by non-machine-learning experts, it is important to provide a solution that enables the whole process to be executed without human intervention;

- **Model validation**. For a newly generated model, it has to be validated before it is deployed into the practical industrial environment. This means that the generated model should be verified, especially with the on-line data, to check the accuracy of the newly generated model. Therefore, it is vital for the reliability and robustness of the platform to ensure the generated model performances before pushing the generated model into the production environment. The transmission errors would result in data errors. Thus, the model validation should be coupled with data validation to address this problem.

- **Seamless model updating**. When a set of new models are generated and validated, it is important to use the new models to replace their corresponding models used in the model serving service. For the industrial environments, many serving processes could not be interrupted. Thus, it is important to design a model seamless updating mechanism.

In addition to the aforementioned requirements, the computing platform needs to process the data near the data source, because enterprises do not want data to be uploaded to the cloud platform from the perspective of security and privacy. This platform that nears the data source can avoid data leakage and reduce transmission delay. However, this platform generally has limited resources, so it should also provide functions related to the general computing demands, e.g., computing resource allocation and isolation. Due to the model complexity and the large scale of data, *trainers'* executions consume considerable CPU, GPU, disk and memory resources. Thus, it important to orchestrate the execution of *trainers* to avoid excessively resource consumption. To achieve maximal flexibility, the existing cloud, fog and edge computing platforms mainly adopt the virtualization techniques, e.g., Hypervisor [30] (system-level virtualization) and container technologies [31] (operating-system-level virtualization). With the maturity of lightweight container technology, platform architectures [24,32] prefer the container techniques such as Docker instead of traditional VM. In the following subsection, a Docker-based service architecture is introduced.

### 3.2. System Architecture

As shown from Figure 1, the computing architecture for CMS is divided into three different layers: the infrastructure layer, the scheduling layer and the model management layer.

In the infrastructure layer, the Docker technology is used for resource allocation and application isolation. Underline resources are divided into dynamic allocable units to be used by various applications. The resource isolation feature of the Linux container enables multiple containers to be executed on a single host without the starving the others. The container orchestration layer focuses on scheduling different types of resources across multiple computers or domains. It can be implemented by reusing an existing Docker manage engine, e.g., Apache® Mesos, Google Kubernetes®, Docker® Swarm, Rancher® Cattle. These two layers are implemented mainly by reusing platforms with industrial big-data-related enhancements, e.g., data accession, data conversion.

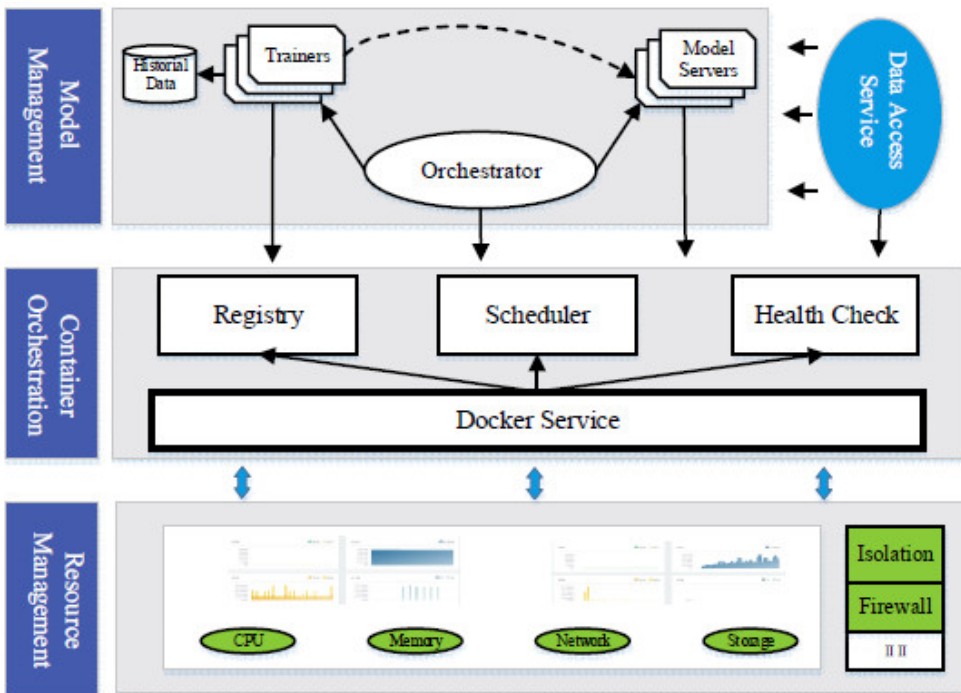

**Figure 1.** System architecture for the industrial application of continuous services.

On top of the container orchestration layer is the model management layer: this layer consists of four core sets of components:

*Data accessors* provide the data access, translation and mapping service from industrial cyber-physical systems. It is also responsible for receiving real-time data from the gateway, translating network protocol, issuing instructions and forwarding data, forwarding processed data to an historical database and real-time database, respectively. Its role also includes the isolation of the traffic between Distributed Control System (DCS) and the big data platform. It can be implemented with an existing data stream processing framework, e.g., Kafka, RabbitMQ, RocketMQ, together with an existing industrial network gateway.

*Trainers* generate models from historical data. Due to the fact of the model evolution, *trainers* have to be executed periodically or triggered by certain events. By taking historical data together with recently updated data as inputs, it emits a newly learned model. Trainers generally need supports from existing machine-learning platforms. A *trainer* normally demands a huge amount of computing resources. Thus, it is of crucial importance to effectively orchestrate *trainers'* executions.

*Modelets* provide the model serving service in production environments. Each time online-time data are received, it queries the models to make prediction or deduce adaptation actions based on its enclosed model. In addition to the model invocation task, one *Modelet* is also responsible for loading, starting, stopping and unloading the model. One major design concern for *modelet* is the strict response time and the supports for scalability.

*Orchestrator* manages the execution of different *trainers* to update their generated models. As can be seen from Figure 1, the *orchestrator* plays a core role in bridging the gap between *trainers* and *modelets*. It triggers the execution of *trainers* to generate up-to-dated models. Another major function of the *orchestrator* is to determine the quality of generated models and to select an appropriated version to switch to. The *orchestrator* needs to seamlessly deploy the selected model to its corresponding *modelet* without interrupting the normal operation.

As one platform might support a dynamic set of *modelets* and *trainers*, it is important to support the dynamicity and scalability of those components. The micro-service pattern is adopted for this architecture implementation. Microservice is a software architecture that decomposes large application systems into a set of independent services; services can be independently developed and deployed,

and realize loosely coupled application development through modular combination [33]. The other important feature is the support for the multitenancy, which is by Google [12], to indicate the scene that multiple model servers and *trainers* deployed in a single instance of the server, due to regulation and cost limitations. The CPU and memory-intensive *trainers* should work together with the *modelets* with stringent time limits, which might lead to cross interference and is challenging to solve. To enhance isolation between operations, this platform provides a set of configuration features to support the resource isolation configuration based on the Docker's resource isolation features. This architecture directly reuses the Docker infrastructure service for the load balance and health checking. The dynamic resource allocation is also implemented with the resource scheduling mechanism provided by the Rancher.

## 4. Orchestrator

An *Orchestrator* contains three major functions—scheduler, model validator and model switcher—which are elaborated in the following subsections.

### 4.1. Scheduler

One important demand for the *trainer* is to update the model periodically to reflect the recent change in monitored or managed equipment. Different *trainers* need to be designed for respective tasks, e.g., metric prediction, classifications, fault identification and optimization. As the model training processes are normally resource intensive tasks, it is important to orchestrate their executions in production environments to lower the peak resource consumption and reduce resource competition.

In order to support the uniform invocation towards different types of *trainers*, this platform adopts the methodology of "Contracts for system design" [34]. A simple *trainer* contract is proposed, which all *trainers* need to follow. This contract includes methods for data acquisition, feature preparation and model training, model version and model exporting. It also defines the format of the generated log file which allows for later model evaluation.

As we can see from Figure 2, a *trainer* contains three major parts: data Extraction, Transformation and Loading (ETL), Training and Model Management. ETL is responsible for reading data from historical databases and performing data extraction and cleaning. After the data are preprocessed, the training interface is invoked by the Orchestrator to perform Feature Engineering (FE) and Algorithm Selection (AS) operations. FE mainly selects relevant features based on the model configuration file, and performs data smoothing, normalization and other feature processing operations. AS uses previously defined machine learning algorithms to train the model. The computation graph is retrained over evolving data. Simultaneously, the scheduler should check the current resource status via the resource monitor before/during scheduling those resource-intensive tasks. It can also assign different resource isolation configuration to balance the conflicting needs between the resource usage efficiency and the task execution efficiency. After the training process, the model-export method is used to export the new model to the model repository.

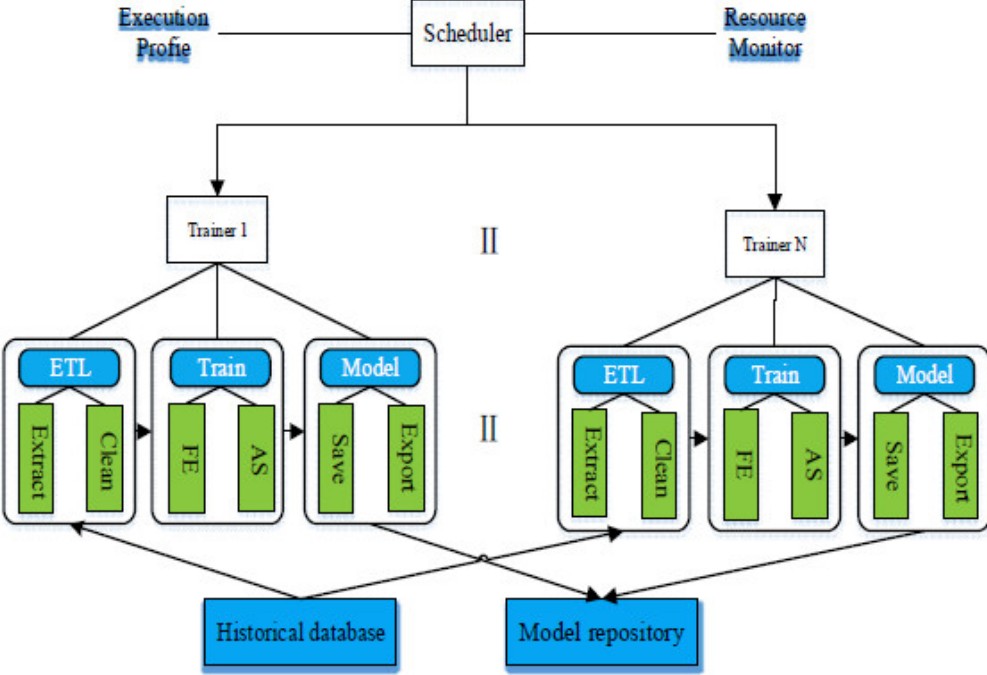

**Figure 2.** Scheduler and the Trainer's interface.

### 4.2. Model Validation

The newly generated model might not always be feasible for production-level usage, due to the disruptions from inconsistent data, broken data, and failures in the underlying execution environments. The broken data might result in generating inaccurate models which cannot be deployed into production environments. Although some data-cleaning techniques can detect and remove errors and inconsistencies from data and improve the data quality, it is difficult to predict whether a learning algorithm will behave reasonably on new data [35]. Thus, the newly generated models should be validated before their usage in production. This is vital to the reliability and robustness of the industrial machine-learning platform, as a bad model might lead to catastrophic consequences.

The model validation contains two different stages: the offline validation and the online validation. The offline validation is performed by checking the prediction quality with the test set when training a new model. Generally, we retain 20% of the data as test data to calculate the MAE (Mean Absolute Error) of the new and old model (the current adopted model) separately. When the new model's MAE is less than the old model, it means that the model verification is passed. The online validation is performed when the model has passed offline validation and is performed on live traffic rather than by the objective function on the training data. In the current prototype, a model would pass the online validation only if running MAE below a certain threshold during an arbitrarily defined period.

### 4.3. Model Seamless Switching Mechanism

After a new model passes the offline validation, it is necessary to replace its corresponding old model. In the *Orchestrator*, a model smooth switching mechanism is proposed to perform the seamless model switching task.

As can be seen from Figure 3, two identical instances are deployed for each model service. Each instance is deployed as a micro-service supported with a respective Docker container. One works as the master and the other works as a hot-backup. For each model invocation, the broker queries both instances. Once the master fails to respond, the broker uses the slave's response. Thus, the model switching process firstly stops the master model instance, loads the updated model and restarts the model serving process. During this process, the broker can still get a model response. After the master node resumes the normal execution, it queries the master instance for the model output. The model

of the slave instance is then updated. During this model switching process, the normal execution of model users can be kept un-touched.

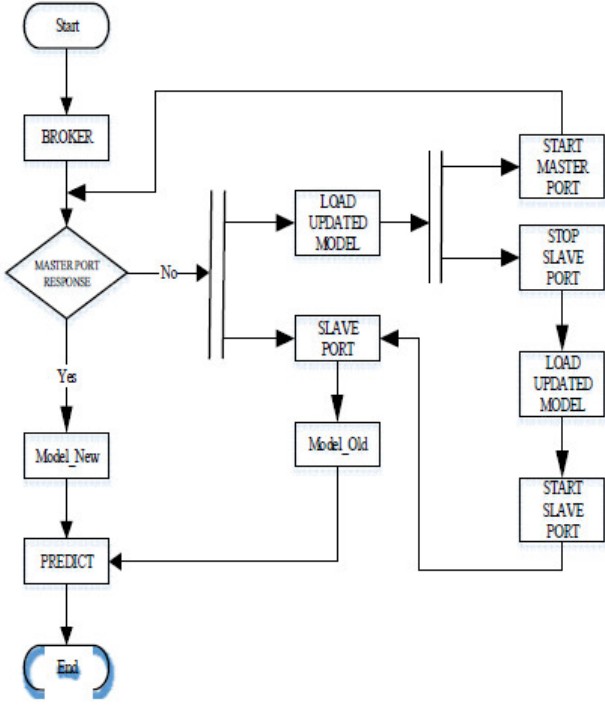

**Figure 3.** The flow chart of the model seamless switching.

## 5. Case Study and Experimental Results

In this section, a case study of a prediction platform for a 1000 MW thermal power plant is introduced and key system metrics are evaluated through several experiments.

### 5.1. Case Study: 1000 MW Thermal Power Plant

One of our deployments of the continuous machine learning platform is the prediction system for 1000 MW coal-fired thermal power plant. This plant has 15,528 physical sensors around different equipment and generates data at about 3.8 G per day. The goal of prediction system is to forecast the future (30 and 60 s) values of major system metrics in a real-time manner. Eight major system metrics, after the discussion with plant operators, are selected for prediction, as shown in Table 1. According to different prediction targets, both experts' experience and data correlation analysis are used to select different features. Around 170 system metrics were selected as the input of models, some of which overleap among different models.

**Table 1.** Information for eight different prediction models.

| Modeling Targets | Features | MAE-30s | Algorithm |
|---|---|---|---|
| Power | 12 | 0.25 | Ridge |
| main steam temperature | 75 | 0.08 | Ridge |
| reheat steam temperature | 93 | 0.11 | Ridge |
| main steam pressure | 48 | 0.16 | Ridge |
| feedwater temperature | 49 | 0.06 | Ridge |
| active power | 10 | 0.20 | Ridge |
| Max superheater wall temp | 10 | 0.06 | LSTM |
| Max reheater wall temp | 10 | 0.05 | LSTM |

The whole processing process starts from the real-time monitored data from the plant's DCS. The data flows this platform through an industrial gateway—HollySys HH800. After the data of the sensors pass through the industrial gateway, they enter Kafka, which is a message middleware. Spark Streaming, a stream processing technique, consumes the data from Kafka and forwards the data to both the real-time database (currently implemented via Redis) and the historical database (currently via MySQL). A *modelet* gets its required features from the real-time database and *trainers* acquire historical data via one custom-designed data conversion package designed to facilitate the data communication between *trainers* and the historical database. An event mechanism is used to synchronize the execution of *modelets*, *trainers* and the *orchestrator*. All those components are implemented as micro-services and deployed as Docker containers. In fact, about 20 different containers are deployed in the system.

*5.2. Model Evolution*

This platform has been deployed into the power plant for 6 months. During this period, this power plant has under a major overhaul, including equipment inspection, cleaning and optimization (from 17 December 2017 to 9 March 2018). The model evolution is clearly observed during this process.

**Impacts from major renovation:** Before the renovation, eight different predictive models are trained with the historical data of recent 1 month. Figure 4a shows the changes in model accuracy in terms of Mean Absolute Percentage Error (MAPE) which is used to indicate the feasibility of the generated model. Before the renovation, all four models can achieve a prediction accuracy under 0.1%, which is rather accurate. However, after the renovation, the continuous usage of existing models might have significant jumps of MAPE, from 4 to 10 times. With the continuous equipment operation, almost all the MAPEs have a rising trend. This clearly indicates that the major renovation has a very important influence on the model. This observation clearly shows the importance of having a model that is periodically updated.

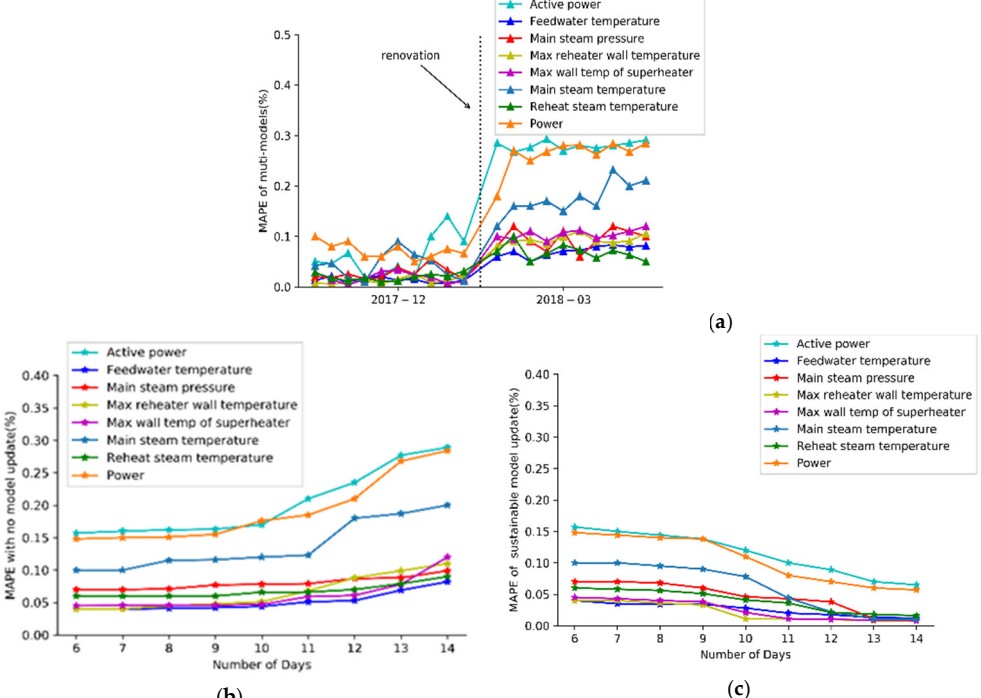

**Figure 4.** Model evolution and effects of model update. (**a**) Model evolution before/after the renovation; (**b**) Mean Absolute Percentage Error (MAPE) without sustainable model update; (**c**) MAPE with sustainable model update.

**Impacts from system adjustment:** The other important observation related to the model evaluation is from the system adjustment. Right after the plant resumed normal operation, the model needs to be updated each day. Otherwise, the model MAPE might also have a significant increase, which is shown in Figure 4b. When normal operation of the plant resumes, a model trained with 5 days' data is used to calculate MAPE for the rest of the 9 days. Figure 4c shows the MAPE trends of the model with a sustainable model update from the sixth days of normal operation after renovation. This fact against our common sense, as the model of the power plant would normally not change at this rate (the original model update cycle is set to 7 days, which is rather enough based on our modeling experience). The plant operators told us that they had been adjusting some important system control parameters during these days. This might explain such fast changes. With our introduced daily model updates (at 1 a.m., a new model is generated with the last 5 days' data), all the MAPEs of the eight models gradually decrease. Thus, it is also important to validate the current execution model in an appropriate way and trigger a model update if the model fails the validation. However, the online validation is challenging as the data distribution of the test data might deviate from train data. How to distinguish the impacts whether from the model deviation or the data deviation remains an area for investigation.

*5.3. Performance Evaluation*

Three different sets of experiments are performed to test the resource usage, impacts of multitenancy and model switching overhead. For the test, CMS is deployed on one Dell Poweredge T620 with e2660v2*2, 144G RAM and two Nvidia 1070 GPU although it can be scaled to different computers.

5.3.1. Resource Usage

The performance of the proposed platform is evaluated from CPU usage, Memory Usage and disk I/O performance, as shown in Table 2. There are two types of container: the continuous execution tasks, e.g., data processors and *modelets*, and intermittent execution tasks, e.g., *trainers*. The data are collected in two different scenes: case with *trainers* off and case with a *trainer* on.

**Table 2.** The resource consumption of key components (the four trainers are off/one of four trainers is on).

|  | Num | CPU (%) | Memory (MB) | Disk (KB/s) |
|---|---|---|---|---|
| Data processors | 10 | 17.24/17.24 | 2824/2825 | 76.6/76.6 |
| Trainers | 4 | 2.12/160.4 | 46.72/ 11059 | 0/176.2 |
| Modelet | 4 | 18.17/18.17 | 369.4/369.8 | 24.2 /24.2 |
| Orchestrator | 1 | 6.3/5.9 | 51.6/48.7 | 17/15.7 |

As shown in Table 2, no matter whether a trainer is on, the resources used by *modelets* and data processors would not change, as these services do not stop during the operation of the platform and their resource consumption is rather limited. The data processors contain Kafka®, Zookeeper®, etc. and the total CPU usage is about 17.24% (100% for one core full usage and this server has about 32 vCores) and about 2825 MB memory as the real-time database is implemented by Redis®, which is a memory database. The message-caching technique of Kafka is also adopted. Those values can be further reduced with lighter configuration, e.g., shortening the data expiry time. The four *modelet* containers total about 369 MB of memory. Their memory usage highly depends on the complexity of the models. In comparison, the *trainers* use more resources; the data shown are is collected when only one of the *trainers* is under execution. However, due to the *orchestrator'* schedules, their peak resource consumption is limited to about 12 GB memory and disk i/o is also within a reasonable range. These results show that this platform is lightweight and can support many *trainers* with limited resources.

5.3.2. Multitenancy with Isolation

The subsection tests the multitenancy performance with different types of resource isolation configurations. A LSTM model *trainer* is used for the superheater temperature prediction and its corresponding *modelet*. This model is updated with comparably small amount of historical data, about 626 MB. The response time of the *modelet* during a *trainer'* execution is collected. This test was conducted in a setting with four vCores and 144 GB memory accessible from the *trainer* and *modelet*. Three different isolation configurations are tested: (1) The complete isolation puts the *trainer* and the model server over the two different sets of vCores (two for each); (2) The no-isolation made no restriction on the CPU vCores. This means that the *trainer* and server would compete for the four vCores. (3) In the partial isolation, the assigned CPU vCores of *trainer* and server overlap with one vCore. Each one is assigned with three vCores.

As shown in Figure 5a, the *trainer* execution starts at about 15 min, indicated by the black dot line. For the complete isolation scene, the training process takes about 18 min and finished at about 33 mins, indicated by the green dashed line. Compared to the stage without training task, the CPU utilization increases about 138%. During the whole simulation period, the prediction delay keeps at about 102 ms, without significant changes. The complete isolation guarantees the exclusive CPU usage, so the prediction response time keeps at about 102 ms during the whole simulation period. With respect to the no-CPU isolation configuration, the prediction and training services are bound to exactly the same four CPU vCores. After the training service is started, the CPU usage increases dramatically to about 194%. This configuration achieves the least training time, about 13 min (the red dotted line). However, due to the CPU competition, the prediction delay was significantly increased by about 300 ms. In a partially isolated environment, the prediction and training services are bound to three CPU vCores with one overleaping. After the training service was started, the CPU occupancy rate was increased by about 180%, and the training time took about 15 min. After the model training is started, due to the duplicated CPU vCore's occupation, the prediction delay increases by about 28 ms. For the memory usage, the training service uses about 10% of memory (about 14 G) in all three scenarios, as no memory isolation is used.

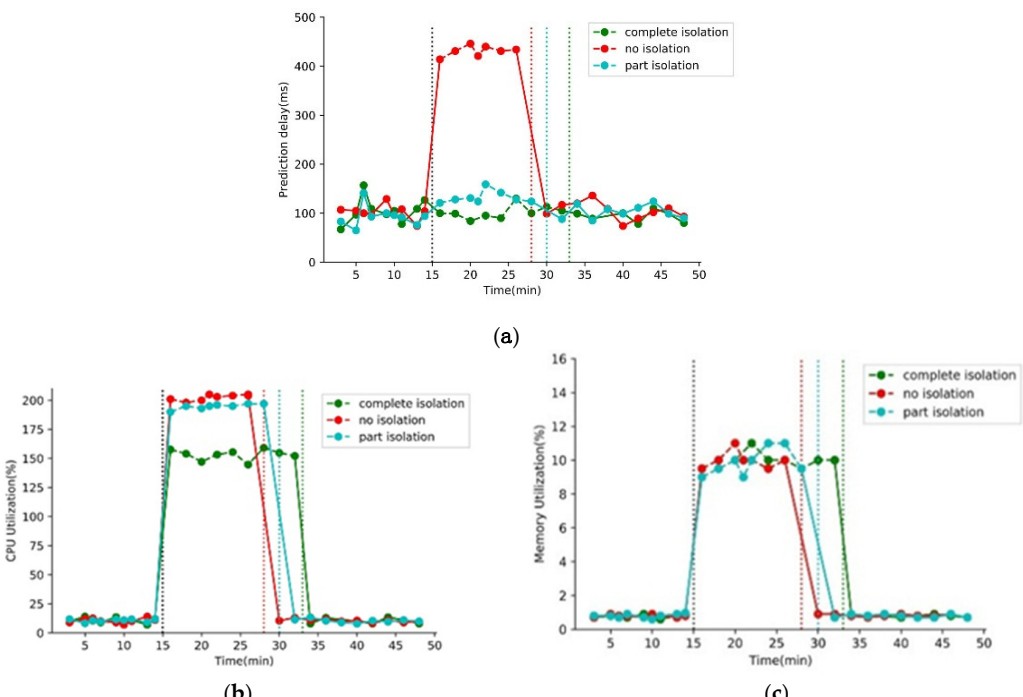

**Figure 5.** Average predictive delay and resource utilization in different isolation environments. (**a**) Prediction delay; (**b**) CPU Utilization; (**c**) Memory Utilization.

As seen from simulation results, although the configuration with no isolation has the highest CPU usage and shortest model training time, it greatly increases the prediction delay. The complete isolation configuration achieves the best protection of the model prediction, however, at the cost of the lowest CPU usage and the longest training time. In comparison, the partial isolation configuration achieves a little increase in prediction time (28 ms) and almost the same CPU usage and training time as the no-isolation configuration. Thus, from this perspective, the partial isolation is the best. However, how to best assign the resources to the different types of services remains a challenge and is one of our future works.

### 5.3.3. Model Switching Overhead

After the newly generated model passed the offline and online validation, the *orchestrator* will replace the old model which is serves as its corresponding *modelet*. Figure 6 presents one *modelet* container load changes and model prediction delay during model switching (one point/3 s). From the proposed master–slave design, when the master port is restarted, the newly generated model is loaded, causing the CPU to rise quickly and form the first peak. Due to the existence of the hot backup model, two models are stored in the memory of the *modelet*. With the release of the memory of the master port model, the *modelet* memory usage drops rapidly and generates a trough. In order to ensure that the master port serving the new model can be predicted normally, we set the interval, restarting the hot backup model to 4s. The reason why the CPU generates the second peak and the corresponding memory trough is due to the restarting of the slave port and the updating and loading of the hot backup model. Although the occupancy rate of the CPU increase, the isolation feature allows the impacts to be limited to within only one core during the entire switching process. The model used for experiment is about 176 KB, which caused Disk to generate 158.6 KB/s extra overhead when the model was switched. However, since the newly generated model already has a cache in memory, the model update of the salve *modelet* does not need to read it again. That is why the disk i/o displays a single peak.

Furthermore, as seen from Figure 6, the prediction delay only observes a slight increase, about 24.2 ms in the whole switching process. Based on master–slave design, the model upgrade process has little effect on the prediction delay, which is the overhead of the service status checking. This service query is rather fast. This design allows for the smoothness and high efficiency of the model switching process.

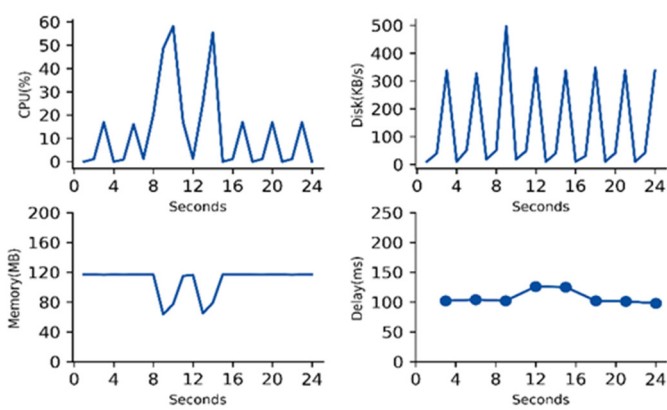

**Figure 6.** Load changes and model prediction delay during model switching.

## 6. Conclusions

In this paper, an industrial machine learning and serving platform for industrial big data is proposed with the support of autonomous model re-training, validating and deployment. This platform provides a clear definition of the component structure of *trainers* and model server to allow consistent component invocation and manipulation, and avoids the usage of ad hoc glue code and custom scripts. The architecture is designed and supported with the container technology. The resource isolation and

scheduling functions, vital for the resource-intensive training tasks and time-contingent serving tasks typical in the industrial environments, are also supported. A model seamless switching mechanism is designed to allow the run-time to update a model without interrupting the normal operation of the model serving process. This platform is implemented with supporting components including, e.g., data streaming, the real-time database, the historical database, typically used in industrial big data. Several machine learning suits, e.g., Spark ML, TensorFlow® and Keras®, are also supported. Based on this prototype, a continuous big data platform is developed for a 1000 MW thermal power station power plant with eight different models and more than 30 updates per model are executed. The advantage of the platform we designed is that it provides industrial model life cycle management and continuous integration services without manual intervention. It can maintain long-term model service quality, help the plant operators make better a decision analysis, and improve factory economic efficiency. Field tests show that this platform achieves good performance under resource consumption, multitenancy with isolation and model updating overheads. The model evolution is also identified and analyzed after more than five months' continuous execution. However, there are still some shortages and areas for improvement on the platform. On the one hand, the update timing of the model service is not dynamic enough to recognize the different industrial conditions and automatically set the update timing. On the other hand, model retraining does not use the old model, and a more efficient retraining method is needed to further reduce the consumption of computing resources.

For future work, we currently focus on the model online validation domain. In industrial environments, it is normal that the training and test data are from the slightly different distribution which is against the assumption typically used in statistical learning theory. Thus, how to effectively identify the model evolution is our current research focus. In fact, research on domain adaptation [36] has been used and some processes have been made and might appear in another paper. The other important field is the resource dynamic allocation and scheduling for the platform to achieve efficient resource usage and scalability.

**Author Contributions:** Conceptualization, K.L. and N.G.; Funding acquisition, N.G.; Methodology, K.L. and N.G.; Project administration, K.L.; Resources, N.G.; Software, K.L.; Supervision, N.G.; Writing—original draft, K.L. and N.G.; Writing—review & editing, N.G. All authors have read and agreed to the published version of the manuscript.

**Funding:** This research was funded by The National Natural Science Foundation of China, grant number 61772473.

**Conflicts of Interest:** The authors declare no conflict of interest.

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
