# Peer review of "CMS: A Continuous Machine-Learning and Serving Platform for Industrial Big Data"

_futureinternet, doi:10.3390/fi12060102_

Round 1

Reviewer 1 Report

  1. The experimental results for case study in section 5 are provided without quality measures (standard error) in the current format. For example, the changes of model accuracy in term of MAPE (mean absolute percentage error) are provided without quality measures. Please add the corresponding standard error for the corresponding estimates in the revised manuscript.
  2. The author should clarify the pros and cons of the proposed method.

Reviewer 2 Report

This paper addresses the topic of providing continuous services. The authors proposed a continuous machine learning and serving platform for industrial big data. The platform was tested in a thermal power plant and results are showing promising results.

The topic is worth researching and relevant to the field of Future Internet Journal. The quality of English also satisfies the level of the journal.

Some things must be addressed before publication.

In the introduction section the sentence "Compared to the cloud computing, the framework that processing data..." must be corrected to "Compared to cloud computing, the framework that processes data..".

The authors should provide a brief description of the micro-service pattern from reference 28 (page 5).

How is the validation process performed? Is the validation data set used or only test and train data sets?

Why are the authors using only data from the last 5 days?

Fig. 3: provide outputs from the "start master port" and "start slave port" steps.

Page 8: "this observation clear shows..." change to " this observation clearly shows..."

What method was used to validate models?

References are not up-to-date, so I suggest adding some more from the recent two years.

Reviewer 3 Report

This paper proposes a container-based CMS platform. The manuscript is well organized. However, it must be carefully reviewed concerning English writing. It is possible to find several errors regarding spelling, grammar, and punctuation (8 errors considering the abstract only).

This reviewer misses some studies of the current state of the art in the introduction, to know:
1) "A comprehensive review on smart decision support systems for health care." IEEE Systems Journal 13.3 (2019): 3536-3545. DOI: 10.1109/JSYST.2018.2890121
2) Industrial Big Data as a result of IoT adoption in manufacturing. Procedia cirp, v. 55, p. 290-295, 2016. DOI: 10.1016/j.procir.2016.07.038
3) Industrial big data analytics and cyber-physical systems for future maintenance & service innovation. Procedia Cirp, v. 38, p. 3-7, 2015.
10.1016/j.procir.2015.08.026

The authors should improve the quality of the figures.

Page 3 - The authors say "The supports for the different machine learning suits mean that the model serving implementations should be designed independently with different model formats." How will the model be able to integrate these different formats?

"The training processes execute the feature selection, feature construction and training process in a defined sequence". How it will be conducted? Explain these processes better.

Fig 1 - In terms of security and privacy, what do the authors suggest? I believe that the authors should describe a little about their vision.

Page 6 - "By comparing the model quality against a fixed threshold and the current adopted model, the model quality can be verified."

How will this comparison take place? What are the suggested thresholds? This is important to be addressed to verify the model quality.

Page 10 - The authors say "As seen from simulation results, although the configuration with no isolation has the highest CPU usage and shortest model training time, it greatly increases the prediction delay." What is the reason for this?

Round 2

Reviewer 3 Report

For this reviewer, the manuscript is ready for publication. The authors addressed all of my suggestions.